# Comparative Analysis Reveals Different Evolutionary Fates and Biological Functions in Wheat Duplicated Genes (*Triticum aestivum* L.)

**DOI:** 10.3390/plants12173021

**Published:** 2023-08-22

**Authors:** Licao Cui, Hao Cheng, Zhe Yang, Chuan Xia, Lichao Zhang, Xiuying Kong

**Affiliations:** 1Institute of Crop Sciences, Chinese Academy of Agricultural Sciences, Beijing 100081, China; cuilicao@jxau.edu.cn (L.C.); 82101219301@caas.cn (H.C.); 82101221605@caas.cn (Z.Y.); xiachuan@caas.cn (C.X.); zhanglichao@caas.cn (L.Z.); 2College of Bioscience and Engineering, Jiangxi Agricultural University, Nanchang 330045, China; 3State Key Laboratory of Crop Stress Biology for Arid Areas, College of Life Sciences, Northwest A&F University, Yangling 712100, China

**Keywords:** wheat, gene duplication, evolutionary rate, domestication, polyploidization

## Abstract

Wheat (*Triticum aestivum* L.) is a staple food crop that provides 20% of total human calorie consumption. Gene duplication has been considered to play an important role in evolution by providing new genetic resources. However, the evolutionary fates and biological functions of the duplicated genes in wheat remain to be elucidated. In this study, the resulting data showed that the duplicated genes evolved faster with shorter gene lengths, higher codon usage bias, lower expression levels, and higher tissue specificity when compared to non-duplicated genes. Our analysis further revealed functions of duplicated genes in various biological processes with significant enrichment to environmental stresses. In addition, duplicated genes derived from dispersed, proximal, tandem, transposed, and whole-genome duplication differed in abundance, evolutionary rate, gene compactness, expression pattern, and genetic diversity. Tandem and proximal duplicates experienced stronger selective pressure and showed a more compact gene structure with diverse expression profiles than other duplication modes. Moreover, genes derived from different duplication modes showed an asymmetrical evolutionary pattern for wheat A, B, and D subgenomes. Several candidate duplication hotspots associated with wheat domestication or polyploidization were characterized as potential targets for wheat molecular breeding. Our comprehensive analysis revealed the evolutionary trajectory of duplicated genes and laid the foundation for future functional studies on wheat.

## 1. Introduction

Gene duplication is an important source of diversity and evolution that provides direct evidence for genetic novelty, adaptation, and speciation [1,2]. The main mechanisms of gene duplication are whole-genome duplication (WGD) and single gene duplication [3]. WGD is a widely common phenomenon in eukaryotes and seems more pervasive in plants than in animals [4]. One or two rounds of genome duplication are reported to precede the angiosperm diversification [5], and only one angiosperm, *Amborella trichopoda*, is known not to have undergone additional WGD [6].

Diploidization is achieved a few million years after WGD through chromosomal rearrangements, gene conversion, gene loss, subgenome dominance, and divergence of expression between duplicate copies, reverting to the disomic inheritance [7,8]. A return to a diploid state quickly follows genome duplication, and various types of single gene duplication occur continuously and commonly throughout the long evolutionary process and are associated with critical environmental adaptations [9,10]. Single gene duplication includes tandem, proximal, transposed, and dispersed duplication [3,11].

Tandem duplication, which arises from unequal crossovers and is often accompanied by inversion events, generates identical genes close to each other on the same chromosome [3]. Proximal gene pairs are close on the chromosome but a few genes apart, and are thought to originate from local transposon activity or ancient tandem duplication events disrupted by the insertion of other genes [12,13]. Transposed duplication could occur by DNA-based or RNA-based transposition (or retro-transposition), producing a duplicated gene relocated to a novel chromosomal locus [11,14]. Dispersed duplication, prevalent in different plant genomes, produces two copies of a gene, neither adjacent nor linked by an unspecified mechanism [15,16]. With the advent of high-quality chromosome-level assemblies, various tools, such as MCscanX [17] and DupGenfinder [18], have been developed to identify the duplication modes in deeply differentiated eukaryotes. These valuable tools laid the foundation for further revealing the gene duplication contributions to morphological complexity, gene regulatory network, and environmental adaptation [18].

Wheat (*Triticum aestivum* L.) has been one of the most important cereal crops globally since the Neolithic Age and offers 20% of proteins and calories consumed by humans in modern society [19]. Extensive cytogenetic, genetic, taxonomical, and phylogenomic studies facilitated the resolution of the origin and evolution of wheat [20]. Wheat is believed to be an allohexaploid species (AABBDD) derived from two successive hybridization rounds within *Triticum* and *Aegilops* genera [21]. The first allotetraploidization, occurring 0.36 to 0.50 million years ago, involved the hybridization between *T. urartu* (AA) and an unconfirmed or extinct species closely related to *Ae. speltoides* (SS) [22], leading to the formation of the allotetraploid wild emmer *T. dicoccoides* (AABB). The primary domestication of wild emmer gave rise to the domesticated emmer (*T. dicoccum*, AABB) with the loss of brittle rachis, followed by the selection of naked or free-threshing kernel tetraploid wheat to generate durum wheat (*T. durum*, AABB) [23]. The second allohexaploidization occurred between the allotetraploid wheat and the diploid *Ae. tauschii* (DD) 8000~10,000 years ago, forming hexaploid wheat [24].

This study aimed to compare different duplication modes of *T. aestivum* in terms of selection pressure, gene structure, expression profiles, biological functions, and nucleotide diversity. Data from this study may also provide candidate genes for further molecular breeding. Thus, the landscape of gene duplication in *T. aestivum* was profiled. Different gene duplication modes were identified for A, B, and D subgenomes, including dispersed duplication, proximal duplication, tandem duplication, transposed duplication, and WGD. The non-synonymous substitution rate (Ka), synonymous substitution rate (Ks), and Ka/Ks ratios were estimated to evaluate the evolutionary rate of different gene duplication modes. The gene compactness, expression profile, codon usage bias, and genetic variation of different duplicated gene types were also compared. The duplicated genes evolved faster, possessed shorter gene lengths, weaker expression, showed higher tissue specificity than non-duplicated genes, and were significantly enriched in various biological processes, especially in response to biotic and abiotic stresses. For duplicated genes, the tandem and proximal duplication-derived genes appear to have undergone more rapid functional divergence than other duplication modes. Moreover, our data demonstrated the asymmetrical patterns between A, B, and D subgenomes in hexaploid wheat. Finally, several duplicated hotspots associated with primary domestication (from *T. dicoccoides* to *T. dicoccum*), secondary domestication or improvement (from *T. dicoccum* to *T. durum*), and polyploidization (from tetraploid wheat to hexaploid wheat, and from *Ae. tauschii* to hexaploid wheat) were identified with publicly available re-sequencing datasets. Overall, our study revealed the evolutionary trajectories of the duplicated genes and laid a substantial foundation for subsequent functional investigations in wheat.

## 2. Results

### 2.1. The Landscape of Gene Duplication in A, B, and D Subgenomes of T. aestivum

The chromosome-level assembly of hexaploid wheat formed the basis for systematically identifying duplication events, particularly for tandem and proximal duplications dependent on the relative chromosomal position [25]. For the *T. aestivum* A subgenome, the DupGen_Finder pipeline identified 2165 genes resulting from WGD, 4923 from tandem duplication, 2482 from proximal duplication, 2126 from transposed duplication, and 5634 from dispersed duplication (Figure 1A, Appendix A). The remaining 18,015 genes not homologous in the outgroup species or involved in any duplication were classified as non-duplicated genes. A similar disparity was observed in *T. aestivum* B and D subgenomes. It is noteworthy that tandem duplication (A subgenome: 28.41%; B subgenome: 30.30%; D subgenome: 30.64%) and dispersed duplication (A subgenome: 32.51%; B subgenome: 31.44%; D subgenome: 32.32%) occupied the majority of the duplicated genes, suggesting that these duplication events occurred more frequently during the long-term evolution and contributed largely to genome duplication in *T. aestivum*.

Chromosomal location analysis showed that WGD tended to be skewed towards the ends of chromosomes with two hotspots at the end of 4A (about 640–745 Mb) and the beginning of 7A (about 0–70 Mb) chromosomes (Figure 1B). The proximal, tandem, and transposed duplication-derived genes were mainly concentrated on both ends of chromosomes, which might be explained by the meiotic homologous recombination in cereals being biased towards the distal end of the chromosome, giving rise to these duplications. Although dispersed duplication-derived genes and non-duplicated genes tended to be located on the chromosome ends, a large proportion of these kinds of duplications were also found to be scattered distribution along the chromosomes.

### 2.2. Genome-Wide Distribution and Correlation Analysis of Ka, Ks, and Ka/Ks

The Ka, Ks, and Ka/Ks were calculated to reveal the evolutionary rate and selection pressure during *T. aestivum* evolution. The distribution of Ka, Ks, and Ka/Ks is more densely located at both ends of the chromosome than in the pericentromeric region, which may be attributed to the lower density of genes in the pericentromeric region (Appendix A). The cor.test revealed that the Ka was positively correlated with the Ks (A subgenome: ρ = 0.40, *p* < 2.20 × 10^−16^; B subgenome: ρ = 0.42, *p* < 2.20 × 10^−16^; D subgenome: ρ = 0.40, *p* < 2.20 × 10^−16^) (Figure 2, Appendix A). Similar results were observed in *A. thaliana* (ρ = 0.21) [26], soybean (ρ = 0.22) [27], and *Pyrus* (ρ = 0.75) [28], suggesting that selection affecting synonymous and non-synonymous loci may be shared across different genomes. We also observed a significant negative correlation between Ks and Ka/Ks and a positive correlation between Ka and Ka/Ks.

It is noteworthy that non-duplicated genes had lower Ka, Ks, and Ka/Ks than duplicates (Ka: 0.0444 vs. 0.0611; Ks: 0.2069 vs. 0.3040; Ka/Ks: 0.2308 vs. 0.2313) (Mann–Whitney U test, *p* < 0.001) (Appendix A), suggesting that non-duplicated genes evolved more slowly and have suffered higher negative selection (Ka/Ks < 1). In addition, a well-documented trend was observed in the different duplication modes, with tandem, proximal, and transposed duplication-derived genes showing qualitatively higher Ka/Ks values, suggesting that these duplication patterns were preserved in *T. aestivum* genome and experienced a more rapid evolution rate than the other categories (Appendix A). By contrast, WGD-derived genes tended to be more conserved with extremely lower Ka/Ks values.

The asymmetrical evolutionary rates were observed for A, B, and D subgenomes. For WGD-derived and dispersed duplication-derived genes, the Ka/Ks values of the *T. aestivum* A subgenome were significantly higher than that of the D subgenomes by using the least significant difference (LSD) test. Moreover, the Ka/Ks values of WGD-derived genes were significantly higher in the *T. aestivum* A subgenome than in the B subgenome. In contrast, the Ka/Ks values of the *T. aestivum* D subgenome showed significantly higher values than the *T. aestivum* B subgenome for transposed duplication-derived genes (Appendix A). No significant difference was observed for proximal duplication, tandem duplication, and non-duplicated genes.

Positive selection represents the process by which a *de novo* advantageous mutation arises and spreads rapidly in a population towards fixation, a process now represented as a hard sweep [29]. Based on Ka/Ks ratios, 193, 133, and 168 positive selected genes were identified for *T. aestivum* A, B, and D subgenomes, providing candidates for further functional investigation in *T. aestivum* (Appendix A). The highest number of positively selected genes was found in non-duplicated genes (A subgenome: 65; B subgenome: 57; D subgenome: 75), followed by dispersed and tandem duplication-derived genes. However, only a few WGD-derived genes (A subgenome: 8; B subgenome: 1; D subgenome: 3) were found to have undergone positive selection.

### 2.3. Comparison between Gene Properties of Different Modes of Duplicated Genes

The rcorr package was employed to perform the pairwise Pearson correlation analysis, results showed that gene length was strongly positively correlated with intron length and moderately positively correlated with exon length, indicating that intron length contributes more to the total gene length (Figure 3A, Appendix A). Additionally, both Ka and Ks were significantly negatively correlated with gene length, exon length, exon number, and intron length. These results suggested a potential relationship between gene structure and evolutionary rate in the *T. aestivum* genome.

Non-duplicated genes tended to possess longer gene length, exon length, and intron length than duplicated genes (gene: 3986.61 vs. 3041.48; exon: 1670.76 vs. 1484.24; intron: 2312.48 vs. 1554.15) (Mann–Whitney U test, *p* < 0.001) (Appendix A), they also had more exons and lower GC content (number of exons: 5.51 vs. 3.89; GC content 56.92% vs. 57.25%) (Mann–Whitney U test, *p* < 0.001). The characteristics of the duplicated genes were assessed to investigate whether duplication modes affect gene structure differentiation. The tandem and proximal duplication-derived genes displayed shorter gene length, exon length, and intron length, whereas the transposed duplication-derived and WGD-derived genes exhibited longer gene compactness (Figure 3B, Appendix A). LSD test was performed to assess the asymmetrical patterns of the gene feature. Apart from transposed duplication-derived genes without significant divergence, the other duplication modes tend to show significant differences between A, B, and D subgenomes (Appendix A). Notably, no asymmetrical difference was observed for the exon number.

### 2.4. Divergence in Codon Usage Bias of Different Modes of Duplicated Genes

Codon degeneracy refers to multiple codons encoding the same amino acid. Codon usage bias is the difference in the relative frequency of synonymous codes in a coding sequence, which has been related to function and adaptation. There are more abundant or preferred codons than others, and fewer rare or non-preferred codons [30,31]. Codon usage bias was evaluated using existing measures, such as the codon adaptation index (CAI) [32], the codon bias index (CBI) [33], and the frequency of optimal codons (Fop) [34], to calculate codon bias based on a set of optimal reference codons from a subset of genes or tRNA concentration data. Pearson correlation analysis revealed that CAI, CBI, and Fop were significantly negatively correlated with Ka and Ka/Ks, but positively correlated with Ks (*p* < 0.001) (Figure 3A, Appendix A). In addition, these three indicators were all significantly negatively correlated with gene length, exon length, intron length, and exon number (*p* < 0.001).

The overall average CAI, CBI, and Fop were 0.2235, 0.0722, and 0.4585 for non-duplicated genes, whereas these three indicators were 0.2323, 0.0949, and 0.4709 for duplicated genes, respectively (Appendix A). Significantly lower CAI, CBI, and Fop were observed for non-duplicated genes compared to duplicated genes (Mann–Whitney U test, *p* < 0.001). Tandem duplication-derived genes exhibited higher codon usage bias among different duplicated modes, whereas transposed duplication-derived genes had lower codon usage bias. We thus speculated that there might be a potential correlation between codon usage bias and duplication patterns in *T. aestivum* (Appendix A). Apart from transposed duplication-derived genes, the remaining modes of duplicated genes showed asymmetrical codon usage between different subgenomes (Appendix A). Meanwhile, there is no significant difference observed for non-duplicated genes.

### 2.5. Expression Levels and Tissue Specificity of Duplicated Genes

It has been previously indicated that a potential relationship between the evolutionary rate and expression profiles (e.g., gene expression level and tissue specificity) [28,35,36]. RNA sequencing has become a powerful tool for gene or isoform expression profiling in biological research [37]. To determine whether the evolutionary rate affects the expression patterns, the publicly available RNA-seq datasets were employed. The expression level and tissue specificity were assessed using the fragments per kilobase of exon per million fragments mapped (FPKM) value and the tissue specificity index (tau or τ). We further explored the correlation between the evolutionary pressure and the expression profile. A significantly negative correlation was found between the Ka/Ks and expression level (Figure 3A, Appendix A). These results are similar to previous studies in *Pyrus* [28] and *Brassica* [36].

The overall expression level of non-duplicated genes (average FPKM = 7.37) was significantly higher than in duplicated genes (average FPKM = 5.85) (Mann–Whitney U test, *p* < 0.001) (Appendix A). On the contrary, a significantly lower tissue specificity was observed for non-duplicated genes (average τ = 0.7599) than in duplicated genes (average τ = 0.7800) (Mann–Whitney U test, *p* < 0.001). Additionally, WGD-derived genes displayed the highest expression level, followed by dispersed duplication-derived genes. Tandem and proximal duplication-derived genes showed extremely low expression levels. Meanwhile, proximal and tandem duplication-derived genes exhibited high tissue specificity, whereas dispersed and transposed duplication-derived genes tend to be expressed in a variety of samples with low tissue specificity (Appendix A). The overall expression levels showed that D (average FPKM = 7.15) > A (average FPKM = 6.54) > B (average FPKM = 6.19). Additionally, the A and D subgenomes showed different expression patterns in response to different duplication modes. For example, the tissue specificity of dispersed, tandem, transposed, and whole-genome duplication-derived genes showed a D > A pattern (Appendix A).

### 2.6. Functional Divergence of Different Types of Duplicated Genes

To understand whether there is a functional divergence between patterns of genes, *T. aestivum* genes were annotated based on rapid orthology using pre-computed eggNOG phylogenies and clusters. Take the A subgenome as an example, the non-duplicated genes were significantly enriched in various plant growth and development processes, such as nitrogen compound metabolic process (GO:0006807), primary metabolic process (GO:0044238), cell cycle (GO:0007049), and cellular component organization or biogenesis (GO:0071840) based on Gene Ontology (GO) enrichment analysis (Figure 4, Appendix A). By contrast, the duplicated genes were shown to play essential roles in stress resistance. For instance, the dispersed duplication-derived genes were significantly enriched in immune response (GO:0006955) and response to chemical (GO:0042221). The proximal duplication-derived genes were the top enriched in response to biotic stimulus (GO:0009607) and response to external stimulus (GO:0009605). The tandem duplication-derived genes were widely involved in response to chemical (GO:0042221), response to abiotic stimulus (GO:0009628), response to stress (GO:0006950), response to external stimulus (GO:0009605), response to biotic stimulus (GO:0009607), and response to endogenous stimulus (GO:0009719). A total of 263 WGD-derived genes were significantly enriched in response to chemical (GO:0042221). In addition, 43 Kyoto Encyclopedia of Genes and Genomes (KEGG) pathways were significantly enriched, whereas the dispersed, proximal, tandem, transposed, and whole-genome duplication-derived genes were significantly enriched in 35, 9, 32, 4, and 10 KEGG pathways, respectively (Appendix A). It is notable that the WGD-derived and dispersed duplication-derived genes were involved in plant-pathogen interaction and environmental adaptation.

We further performed the GO and KEGG enrichment analysis for the B and D subgenomes of *T. aestivum* (Appendix A). Similar to A subgenome, the non-duplicated genes were enriched in various metabolic and biogenesis processes, whereas the duplicated genes were significantly enriched in various biotic and abiotic processes. Compared to the D subgenome, the tandem duplication-derived gene of the A and B subgenomes were additionally enriched in aging (GO:0007568) and seed germination (GO:0009845). Moreover, for proximal duplication-derived genes, the B subgenome was specifically enriched in the system process (GO:0003008) and the biological process involved in symbiotic interaction (GO:0044403). The D subgenome was shown to be specifically enriched in aging (GO:0007568) and response to stress (GO:0006950). By contrast, there are no specific enriched GO terms for the A subgenome. A similar phenomenon was observed for the other duplicated modes. These results suggested that genes with the same fate may have similar biological functions. Meanwhile, the duplicated genes can undergo neo-functionalization, sub-functionalization, or non-functionalization, resulting in the asymmetrical biological function of A, B, and D subgenomes in *T. aestivum*.

### 2.7. The Contribution of Duplicated Genes to the Expansion of Gene Families

To conduct a comparative genomics analysis, we gathered protein-coding sequences from a total of 19 genomes or subgenomes belonging to 15 species within the genera *Triticum* and *Aegilops*. Specifically, within the *Triticum* genus, *T. dicoccoides* was divided into *T. dicoccoides* A and B subgenomes, *T. durum* was divided into *T. durum* A and B subgenomes, and *T. aestivum* was divided into *T. aestivum* A, B, and D subgenomes. We identified 48,022 orthologous groups, with 8811 orthologous groups shared by all genomes or subgenomes (Appendix A). Single-copy orthologous groups dominated the different genomes, and their proportion decreased as the number of ortholog members increased (Figure 5B). Furthermore, 28 *T. aestivum* A-specific genes, 24 *T. aestivum* B-specific genes, and 15 *T. aestivum* D-specific genes were assigned to 12, 10, and 5 orthologous groups, respectively.

To infer the phylogenetic position of *T. aestivum*, the protein sequences of 3313 single-copy genes were used to construct the species tree (Figure 5A). The tree was rooted with *H. vulgare* and *H. spontaneum* as outgroups. Three major taxa corresponding to wheat A, B, and D lineages were identified. Of these, *T. aestivum* A subgenome and its ancestors (*T. urartu*, *T. dicoccoides* A, and *T. durum* A) formed A lineage. *T. aestivum* B subgenome, its ancestors (*T. dicoccoides* B and *T. durum* B), and *Ae. speltoides* were grouped into B lineage. The remaining D lineage contains a subclade consisting of *T. aestivum* D and *Ae. tauschii*, and four species for the Sitopsis section, including *Ae. longissima*, *Ae. sharonensis*, *Ae. bicornis*, and *Ae. searsii*. In addition, *H. marinum*, *S. cereale*, and *Th. elongatum* were grouped separately from the *T. aestivum* A, B, and D lineages, indicating earlier genetic differentiation. These results were consistent with the phylogenetic relationships described in previous studies [21,38,39].

The gene family expansion and contraction patterns in *T. aestivum* were evaluated based on the maximum likelihood modeling of gene gain and loss. The numbers of expanded and contracted families are presented along the phylogeny branches (Figure 5A). A total of 2060, 2068, and 1418 gene families were expanded in *T. aestivum* A, B, and D subgenomes, whereas 6750, 4321, and 4580 were contracted for these three subgenomes, respectively. The proportion of non-duplicated genes (*T. aestivum* A: 1.25%; *T. aestivum* B: 1.56%; *T. aestivum* D: 0.52%) involved in gene family expansion was much lower than that of duplicated genes (Figure 5C). The expanded gene families were derived mainly from the dispersed duplicated genes (*T. aestivum* A: 14.80%; *T. aestivum* B: 14.67%; *T. aestivum* D: 9.50%). Comparatively, the proportion of genes duplicated in *T. aestivum* A and *T. aestivum* B subgenomes involved in gene family expansions was higher than in *T. aestivum* D subgenomes. Specifically, the contribution of tandem duplication and proximal duplication events to gene family expansion was more than twice as high in subgenomes A and B than in subgenome D.

### 2.8. Nucleotide Variation and Genetic Bottleneck in the Domestication and Polyploidization of T. aestivum

To explore the full spectrum of variation in the *T. aestivum* genome, the variant call format files from publicly available databases were downloaded to present SNPs. Approximately 1.25, 1.61, and 0.55 million SNPs were identified in the genic regions (excluding the upstream and downstream regions of genes) of the *T. aestivum* A, B, and D subgenomes, respectively (Appendix A). The transition-to-transversion ratio (Ts/Tv) was 1.85, with C-to-T (17.21%) and G-to-A (17.20%) occupying the most abundant allelic substitution categories (Appendix A), suggesting that there are fewer purine-to-purine or pyrimidine-to-pyrimidine mutations than pyrimidine-to-purine or purine-to-pyrimidine mutations in the genome of *T. aestivum*. Over half of the nucleotide variants were located in the intron region, followed by synonymous and non-synonymous mutations in the exon region (Figure 6A, Appendix A). In addition, about 8.5% of the variants occurred in the 3′ UTR and 3% in the 5′ UTR.

We assessed the effect of genetic bottlenecks (measured by reduced nucleotide diversity) at the population level for the following three processes during the evolutionary history of *T. aestivum*, including primary domestication (*T. dicoccoides* vs. *T. dicoccum*), secondary domestication or improvement (*T. dicoccum* vs. *T. durum*) and polyploidization (Tetraploid wheat vs. Hexaploid wheat and *Ae. tauschii* vs. Hexaploid wheat) [21,40,41]. The heterogeneity of gene length makes it difficult to evaluate individual genes with fixed sliding windows; we thus calculated the site-pi values for each locus of variation and then averaged them on a gene-by-gene basis. During primary domestication, the genetic diversity of the A and B subgenomes decreased by 24.74% and 28.04%. In comparison, the genetic bottleneck was 15.46% and 22.72% from *T. dicoccum* to *T. durum*, suggesting the difference in the selection pressure between different subgenomes of *T. aestivum* during artificial selection (Figure 6B,C).

During polyploidization, *T. aestivum* A and B subgenomes lost 43.07% and 40.72% of genetic diversity, respectively. By contrast, the *T. aestivum* D subgenome showed a significant reduction with 72.69% genetic loss (Figure 6B–D). The extremely low genetic diversity of the *T. aestivum* D subgenome compared to the A and B subgenomes can be explained by the extreme genetic bottleneck caused by the wheat hexaploidization process involving a few *T. aestivum* D ancestral individuals [42]. However, interspecific introgression, especially from wild emmer, was the main reason for the relatively high diversity in *T. aestivum* A and B subgenomes [41,43]. Our results also demonstrated that the polyploidization of *T. aestivum* was subject to stronger genetic pressure than the artificial selection processes of domestication and improvement.

Divergent patterns of genetic bottleneck were also observed for different duplication modes (Appendix A). For example, tandem duplication-derived genes on the A subgenome endured the strongest genetic bottleneck (26.35%) during domestication, while the transposed duplication-derived genes on the B subgenome were lost the most (29.92%). In contrast, dispersed duplication-derived genes experienced the most severe genetic reduction during the improvement process (A: 19.74%; B: 24.41%).

We further calculated the pairwise linkage disequilibrium within each gene. As expected, artificial selection reduces genetic diversity and elevates linkage disequilibrium [44,45]. The linkage disequilibrium values increased dramatically from *T. dicoccoides* (0.7178 for the A subgenome, 0.7599 for the B subgenome) to *T. dicoccum* (0.8103 for the A subgenome, 0.8376 for the B subgenome), and *T. durum* (0.8388 for A subgenome, 0.8429 for the B subgenome). Our results also demonstrated that genes arising from different duplication models diverged in their linkage disequilibrium values (Appendix A).

### 2.9. Hotspots of Duplicated Genes during Wheat Domestication and Hexaploidization

To understand the selective sweep during wheat domestication and hexaploidization, the genome-wide nucleotide diversity (π) and fixation index (F_st_) were measured using an empirical approach with fixed window sizes and window steps. Given that a more stringent filtering criterion was used to capture variation, the determination of selective elimination intervals might filter out some genomics intervals [21,46]. The top 5% windows of the selected fixation index (F_st_ ≥ 0.5205) and π ratio (θπ ≥ 6.3233) values were identified as domestication-related signals. Consequently, 145.2 Mb genomic regions (1.44% of the A and B subgenomes, containing 704 genes) showed significant differences between *T. dicoccoides* and *T. dicoccum* (Figure 7A, Appendix A). Specifically, 347 and 357 selected genes were in the A and B subgenomes, respectively. The selected genes tended to be biased toward distribution on chromosomes 5B (126 genes), 3A (124 genes), and 4A (98 genes). Moreover, 29 Mb (0.29% of the A and B subgenomes) harbored 233 genes and was determined as improvement-related candidates (Figure 7B, Appendix A). Artificial selection for the improved region was enriched on chromosomes 5B (86 genes) and 2A (53 genes). During wheat hexaploidization, we identified 82.7 Mb (599 genes), 26.5 Mb (148 genes), and 40.3 Mb (287 genes) regions as selective sweeps for A, B, and D subgenomes, respectively (Figure 7C,D, Appendix A).

Transcription factor activates or represses transcription of a target gene through binding to specific DNA elements, and plays essential roles in multiple biological processes [47]. Forty-four (*T. dicoccoides* vs. *T. dicoccum*), 6 (*T. dicoccum* vs. *T. durum*), 39 (Tetraploid wheat vs. Hexaploid wheat), and 26 (*Ae. tauschii* vs. Hexaploid wheat) selected genes were identified as transcription factors (Appendix A). We identified a hotspot (2B: 82.361–83.956 Mb) within the domestication-related selective sweep region, which contains four NAC genes formed by tandem duplication. During domestication, their genetic diversity was significantly reduced. Although these genes showed high tissue specificity (τ > 0.75), divergence in expression patterns occurred between members (Figure 8A,B). *TraesCS2B02G118300* was barely expressed at all periods. *TraesCS2B02G118200* was also expressed at lower levels, except for E2 (embryo, pre-embryo stage), E8 (embryo, mature embryo stage), GR5 (grain, 5 days post-anthesis stage), and GR10 (grain, 10 days post-anthesis stage) stages with FPKM values slightly larger than 1. *TraesCS2B02G118400* and *TraesCS2B02G118500* were more highly expressed in the late stage of post-anthesis, and the leaf and root of five-leaf-stage seedlings compared to other tissues or stages. In addition, we identified a compartment on chromosome 5A formed by tandem duplication that included three B3 genes. Among them, no variation was found within the genic region of *TraesCS5A02G438500*, whereas *TraesCS5A02G438600* and *TraesCS5A02G438700* suffered extensively genetic bottleneck during polyploidization, losing 98.46% and 92.05% of their genetic diversity, respectively (Figure 8C). *TraesCS5A02G438700* had an extremely high tissue specificity of 0.9947 and was specifically expressed in the leaf late embryo stage. In comparison, *TraesCS5A02G438600* was expressed throughout the embryonic stage, with peak expression levels in the leaf of the late embryo stage. The 410.539 Mb to 411.226 Mb region on chromosome 5D covered a total of 11 ERF genes formed by five proximal and six tandem duplicated ERF genes, except for *TraesCS5D02G317200*, *TraesCS5D02G317400*, *TraesCS5D02G318300*, and *TraesCS5D02G318500* which are highly expressed in the leaf, stem, and root of five-leaf-stage seedling. The remaining members showed extremely low and no expression patterns at various tissues or stages.

In addition, we identified a series of dispersed duplication-derived genes that were selected during the evolution from *T. dicoccum* to *T. durum*. *TraesCS2A02G338200* (MYB) and *TraesCS2A02G338300* (NAC), and *TraesCS6A02G181400* (ERF) tended to express in various tissues or stages, such as endosperm and booting stage. *TraesCS4B02G037000* (C3H) was expressed throughout the post-anthesis stages with relatively low tissue specificity, whereas *TraesCS6B02G167100* (ARF) was mainly expressed in the embryo, and spike from the booting and heading stages. In brief, these results provide candidates for further molecular cloning and breeding application for wheat and other cereal crops.

## 3. Discussion

### 3.1. Evolutionary Rate, Gene Compactness, and Expression Patterns between Duplicated Genes and Non-Duplicated Genes

Selective pressures can be important in determining the evolutionary fate of duplicated genes [48,49]. Deciphering gene duplication and the evolutionary fate of genes after duplication forms the basis for understanding the plant genome. This study identified 17,330, 17,868, and 16,644 duplicated genes for *T. aestivum* A, B, and D subgenomes, respectively (Figure 1A, Appendix A). The Ka, Ks, and Ka/Ks values were calculated for each homologous pair to evaluate their evolutionary rate. The average values of Ka, Ks, and Ka/Ks for duplicated genes were larger than those for non-duplicated genes, suggesting that duplicated genes evolved more rapidly than non-duplicated genes in *T. aestivum* [50,51].

The relationship between evolutionary rate and gene structure has been documented [52,53]. However, there is a lack of relevant research on crops, especially polyploid wheat. In addition, the decisive relationships between evolutionary rates, gene expression, symbiotic preferences, and biological functions have been poorly understood. In our study, duplicated genes underwent strong evolutionary pressure during evolution, which might be correlated with the compactness of their genes (Appendix A). The gene length, intron length, and exon length of duplicated genes were shorter than those of non-duplicated genes, and the duplicated genes possessed lower exon numbers and higher GC content. The expression levels and tissue specificity of duplicated genes and non-duplicated genes were also investigated based on RNA-seq data. The duplicated genes exhibited lower expression levels and higher tissue specificity patterns. We, therefore, hypothesize that a decrease in expression levels may alter functional redundancy following gene duplication. The CAI, CBI, and Fop values of duplicated genes were higher when compared to non-duplicated genes. In addition, functional enrichment analysis showed that non-duplicated genes were enriched in various plant growth and developmental processes, while duplicated genes were significantly enriched in biotic and abiotic stresses. These results suggest that evolutionary pressure might have operated in various species, and selection pressure may be potentially relevant to gene structure, expression profile, and biological function.

### 3.2. Divergence in Evolutionary Rate, Expression, and Function between Duplication Modes

Since the previously used genome did not reach the chromosome level, severely skewed results were obtained for position-dependent duplication models, such as tandem and proximal duplications [18]. The latest genome thus provides valuable information for systematically revealing the landscape of gene duplication in *T. aestivum* [25]. Different modes of gene duplication exhibited divergent patterns of evolutionary rate and functional evolution. Our results demonstrated that the Ks peaks of WGD, transposed duplication, and dispersed duplication tended to be overlapped (Figure 2B, Appendix A), suggesting that WGD events were accompanied by extensive duplications of transposed and dispersed-derived genes. Compared to WGD-derived duplicates that evolved more conserved, the tandem and proximal duplication-derived genes tend to display qualitatively higher Ka/Ks ratios. This finding demonstrated that preserved younger tandem and proximal duplications underwent more rapid sequence divergence than other duplicated modes, although co-evolution might also have preserved a greater degree of homogeneity in tandem or proximal duplication-derived genes than in genes that are not in the same location as each other [54,55]. The proximal and tandem duplication-derived genes seemed to suffer faster functional divergence, indicating that the positive selection might play essential roles in the initial stage of duplicated gene retention [18]. Expression divergence and biological function between duplicated genes were also investigated. Physically linked duplications, including tandem and proximal duplications, showed lower expression levels, similar to many prior studies [56,57]. We thus speculate that physically linked genes in the same syntenic block are preferentially retained in *cis*-PPIs (protein-protein interactions) after WGD [58], and gene redundancy results in low expression of genes with similar functions.

### 3.3. The Asymmetrical Evolutionary Patterns of the A, B, and D Subgenomes in T. aestivum

Polyploidization is widespread throughout flowering plants, profoundly affecting genome complexity, and generating beneficial genetic diversity for the adaptive evolution of plants [59]. Compared to its diploid and tetraploid ancestors, *T. aestivum* shows significant genomic plasticity and wider adaptability, partly attributed to the generation of new genetic diversity following allopolyploidization [60]. After polyploid formation, the A, B, and D subgenomes were subjected to asymmetrical homoeologous expression [61], epigenetic regulation [62,63], nucleotide variation, and selection [64]. A similar result was observed in *Brassica oleracea*, its draft genome revealed numerous chromosome rearrangements, divergence in gene expression, asymmetrical amplification of transposable elements, and tandem duplications [59]. In addition, structural variation provides evidence for asymmetrical subgenomic evolution and homologous expression divergence in tetraploid peanuts. Genes associated with structural variation are affected by natural selection and human domestication and may influence agronomic traits such as pod size and development [65]. In cotton, asymmetrical subgenome evolution was involved in fiber traits [66]. To better reveal the asymmetrical evolutionary patterns of duplicated genes in *T. aestivum*, the bioinformatics workflow was employed to evaluate the subgenomes divergence by integrating multi-omics data. Our results demonstrated the asymmetrical pattern in evolutionary rate, gene properties, codon usage bias, tissue specificity pattern, and genetic variation of different duplicated genes in *T. aestivum* (Appendix A). In terms of gene features and codon preferences, transposed duplication-derived genes do not tend to differ significantly between A, B, and D subgenomes. However, the asymmetry between subgenomes was observed for genes of other duplication modes, such as proximal and tandem duplication-derived genes, implying that genes with different duplication modes differ in their subgenomic evolutionary history. It is noteworthy that the D subgenome has significantly less genetic diversity compared to the A and B subgenomes. The diversity recovery model in *T. aestivum* proposes that *T. aestivum* has evolved with both severe genetic bottlenecks such as domestication, improvement, and polyploidization, which reduced the genetic diversity of *T. aestivum*, and a large amount of genetic introgression of tetraploid wheat into common wheat to increase the genetic diversity of *T. aestivum*, resulting in asymmetrical subgenomes A, B, and D with different genetic diversity [21]. There are also subgenomic differences in the genetic bottlenecks endured by genes originating from different duplication events during domestication, improvement, and polyploidization. We, therefore, hypothesized that the rich genetic resources provided by gene duplication have provided a wealth of innovative and valuable material for crop evolution and artificial selection. In summary, patterns of asymmetrical duplications in *T. aestivum* will shed light on our understanding of *T. aestivum* genome evolution and provide genomic tools for agronomic traits in polyploid crops of global importance for economic and food security.

### 3.4. Duplicated Genes Might Play Essential Roles in Various Plant Growth and Stress Response of T. aestivum

Gene duplication provides a vast reservoir of novel genes for innovation in functional and phenotypic traits and is believed as a major force driving genome evolution in flowering plants [67,68]. GO and KEGG revealed the potential biological functions of duplicated genes in *T. aestivum* (Appendix A). The GO terms, such as biosynthetic process (GO:0009058), secondary metabolic process (GO:0019748), regulation of biological process (GO:0050789), response to biotic stimulus (GO:0009607), response to chemical (GO:0042221), and response to endogenous stimulus (GO:0009719), and the KEGG pathways, such as oxidative phosphorylation, plant-pathogen interaction, biosynthesis of other secondary metabolites, and photosynthesis were top enriched. These results indicated that these duplicated genes were involved in various biosynthesis and metabolic processes, especially in response to multiple stresses.

The evolutionary fate of duplicated genes depended largely on their potential to evolve new functions [69]. In general, the predominant fate of duplicated genes was silencing or loss due to redundancy of gene function, termed pseudogenization or non-functionalization. However, retained duplicates provide a rich source of evolutionary novelty and biological complexity, including developing complementary gene functions through sub-functionalization, evolving new functions via neo-functionalization, or retention in complex regulatory networks with differential gene expression due to dosage effects [70,71]. To date, our understanding of the potential functions of duplicated genes can be largely guided by studies using transcriptomic data. The spatiotemporal expression patterns of the genes in various tissues or stages suggest that the duplicated genes may also play essential roles in *T. aestivum* growth and development. We focused on several hotspots of duplicated genes during wheat domestication or hexaploidization. These genes within the same syntenic block showed strong tissue specificity with different expression patterns, suggesting the potential sub-functionalization, neo-functionalization, or pseudogenization of these candidates. The analysis of nucleotide variants additionally demonstrated that the diversity levels of these candidate genes underwent significant artificial selection as a result of domestication or hexaploidization, suggesting that these processes facilitate the preservation of particular haplotypes that were absent prior to selection. Characterizing the domestication history of duplicated genes will provide new insights into the differences in functional genes of *T. aestivum* and will help establish links between genetic variation and important agronomic traits. In conclusion, genomic resources are uniquely valuable for the study of crop domestication, polyploid genome evolution, and genomic-assisted improvement of wheat production.

## 4. Materials and Methods

### 4.1. Identification of Gene Duplication Modes in T. aestivum

The genome assembly IWGSC RefSeq v1.0 was downloaded from https://urgi.versailles.inra.fr/download/iwgsc/IWGSC_RefSeq_Assemblies/v1.0/ (accessed on 1 April 2023). Gene annotation IWGSC RefSeq v1.1 was accessible at https://urgi.versailles.inra.fr/download/iwgsc/IWGSC_RefSeq_Annotations/v1.1/ (accessed on 1 April 2023). The genome assembly of *H. vulgare* Morex V3 was retrieved from http://doi.org/10.5447/ipk/2021/3 (accessed on 1 April 2023) and was used as the outgroup species. The mode of gene duplication was identified using the DupGen_finder tools. Notably, the DupGen_finder-unique was a more stringent version of DupGen_finder, designed to eliminate redundant duplicates in various modes [18]. In cases where the same gene was assigned to different gene duplication modes, the modes were categorized as unique based on the following priority order: WGD > tandem duplication > proximal duplication > transposed duplication > dispersed duplication.

### 4.2. Evolutionary Rate Calculation

The InParanoid v8.0 software was employed to identify orthologs between *T. aestivum* and *H. vulgare* [72]. Multiple sequence alignment of proteins was conducted using Clustal Omega v1.2.4 [73]. The alignment of amino acids and their corresponding mRNA sequences was then converted to codon alignment using PAL2NAL (http://www.bork.embl.de/pal2nal/) (accessed on 10 April 2023) [74]. The calculation of Ka, Ks, and Ka/Ks was performed using the codeml package in PAML v4.9 [75]. Orthologous gene pairs exhibiting Ka > 2, Ks > 2, and Ka/Ks > 10 were excluded from subsequent analyses due to the presence of saturated synonymous substitutions and potential inaccuracies in the calculation [76]. For *T. aestivum* genes with one-to-many or many-to-many orthologous gene clusters, an average was computed. Generally, values of Ka/Ks < 1, =1, and >1 indicated purifying selection, neutral selection, and positive selection, respectively.

### 4.3. Codon Usage Bias and Gene Compactness Analysis

To assess the bias in codon usage, the coding sequences were subjected to a series of filtering criteria. These criteria included the presence of a start codon (ATG) at the beginning and a stop codon (TAA, TAG, or TGA) at the end, a minimum length of 300 base pairs, and a length that was a multiple of three. The codon usage indices, such as CAI, CBI, and Fop, were determined using CodonW v1.4.4 (http://codonw.sourceforge.net/) (accessed on 10 April 2023). A custom Python script was developed to determine gene length, mRNA length, exon length, first exon length, intron length, exon number, and GC content based on the generic feature format file.

### 4.4. Enrichment Analysis of Different Modes of Duplicated Genes

The GO and KEGG annotations were performed using EggNOG-mapper v2.1.7 (http://eggnog-mapper.embl.de/) (accessed on 10 April 2023). Subsequently, the GO and KEGG enrichment analysis was conducted using TBtools v1.120 [77]. The GO terms and KEGG pathways that exhibited both *p* < 0.05 and corrected *p* < 0.05 were considered statistically significant. The visualization of GO terms was set at levels 2 and 3. The prediction of transcription factor families was performed using the Plant Transcription Factor Database v5.0 (http://planttfdb.gao-lab.org/prediction.php) (accessed on 10 April 2023) [78].

### 4.5. Gene Family Expansion and Contraction Analysis

Orthologous gene families from *T. aestivum* and other genomes, including *Ae. bicornis*, *Ae. longissima*, *Ae. searsii*, *Ae. speltoides*, *Ae. tauschii*, *H. marinum*, *H. spontaneum*, *H. vulgare*, *S. cereale*, *T. dicoccoides*, *T. durum*, *T. urartu*, and *Th. elongatum* were obtained using OrthoFinder v2.5.4 with the parameters “S diamond -M msa” [79]. The polyploid genome was divided into diploid genomes, such as *T. dicoccoides* A and *T. dicoccoides* B subgenomes, respectively.

The r8s software was utilized to generate an ultra-metric tree by employing 3313 single-copy orthologs [80]. The calibration time was established at a median time of 10.3 million years ago, specifically between *H. vulgare* and *Ae. tauschii*, by querying the TimeTree database (http://timetree.org) (accessed on 20 April 2023) [81]. The CAFÉ v4.2 was employed to ascertain gene family expansion and contraction by utilizing default parameters [82].

### 4.6. Expression Profile Analysis

A total of 80 spatiotemporal RNA-seq samples (BioProjects: PRJNA497810, PRJNA532455, and PRJNA525250) of *T. aestivum* were acquired from the National Center for Biotechnology Information (NCBI) Sequence Read Archive (SRA) database (https://www.ncbi.nlm.nih.gov/) (accessed on 5 April 2023). Detailed information can be found in Appendix A. Quality control was implemented using Trimmomatic v0.36 [83]. The genomic index was constructed and the high-quality reads were aligned to the reference genome of wheat using Hisat2 v2.1.0 [84]. The BAM files were sorted using SAMtools v1.3.1 [85]. The expression levels of each gene were quantified by calculating FPKM with the aid of Stringtie v1.3.5 [86]. The log2-transformed FPKM+1 values were then visualized using the pheatmap package in the R statistical environment. The tissue specificity index τ was utilized to assess the degree of tissue specificity, ranging from 0 to 1. A lower value indicates lower tissue specificity, while a higher value suggests higher tissue specificity [87].

### 4.7. Nucleotide Diversity, Phylogenetic Relationships, and Selective Sweep Analysis

The updated genome-wide genetic variation map of the genera *Triticum* and *Aegilops* was obtained from the Genome Variation Map database (accession number GVM000272) (https://bigd.big.ac.cn/gvm) (accessed on 5 April 2023). Nucleotide diversity and fixation index were computed using the VCFtools v0.1.17 genome toolbox [88]. The pairwise linkage disequilibrium coefficient was calculated using VCFtools and PLINK v1.90b6.21 [89].

### 4.8. Plotting and Statistical Tests

Plotting and statistical analyses were performed in the R language environment. The scatter, box, density, histogram, and stacked plots were generated using the ggplot2 package. The correlation matrix was visualized using the corrplot package. The Mann–Whitney U test, Spearman’s rank correlation test, and LSD test were carried out using the wilcox.test, cor.test, and LSD.test packages, respectively. Statistical significance was determined at the levels of * *p* < 0.05, ** *p* < 0.01, and *** *p* < 0.001.

## Figures and Tables

**Figure 1 plants-12-03021-f001:**
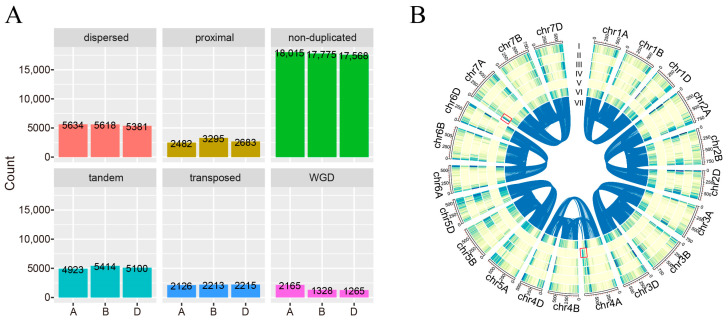
The landscape of gene duplication in *T. aestivum*. (**A**). The gene number of different duplication modes. (**B**). Circos plot of the duplicated genes. From outer to inner circles of the Circos plot represent dispersed duplication (I), proximal duplication (II), tandem duplication (III), transposed duplication (IV), whole-genome duplication (V), non-duplicated genes (VI), and syntenic relationships (VII). Red boxes at the end of chromosome 4A and the beginning of chromosome 7A indicate the hotspot regions of dispersed duplication.

**Figure 2 plants-12-03021-f002:**
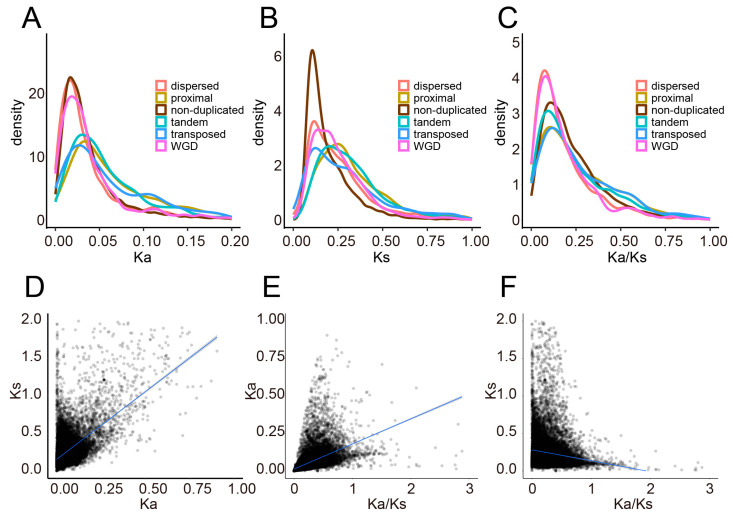
The distribution and correlation analysis of Ka, Ks, and Ka/Ks for the A subgenome of *T. aestivum*. (**A**–**C**). The density plot of Ka, Ks, and Ka/Ks for the A subgenome of *T. aestivum* (**D**). The correlation analysis between Ka and Ks. (**E**). The correlation analysis between Ka/Ks and Ka. (**F**). The correlation analysis between Ka/Ks and Ks.

**Figure 3 plants-12-03021-f003:**
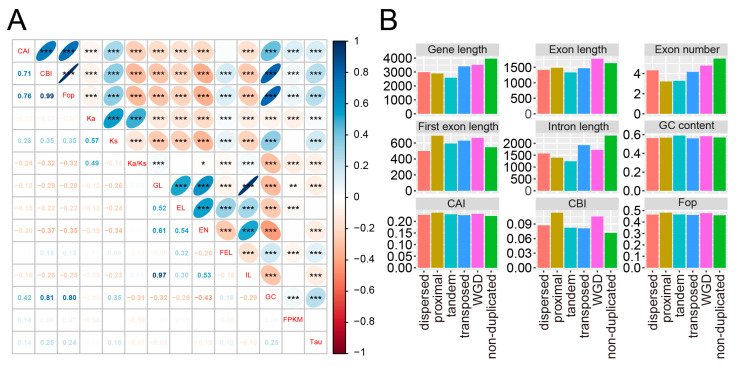
The correlation and statistics of various parameters for the A subgenome of *T. aestivum* (**A**). Correlations among evolutionary rate, gene feature, codon usage bias, and expression profile. The size of the circles in the upper right corner of the figure represents the magnitude of the correlation coefficient, with red indicating a negative correlation and blue a positive correlation. The ribbon on the right represents the correspondence between color and correlation. One asterisk (*), two asterisks (**), and three asterisks (***) indicate a significant difference level of 0.05, 0.01, and 0.001, respectively. The bottom left represents the correlation coefficient values. GL: Gene length; EL: Exon length; EN: exon number; FEL: First exon length; IL: Intron length. (**B**). Statistics of parameters in different duplication modes.

**Figure 4 plants-12-03021-f004:**
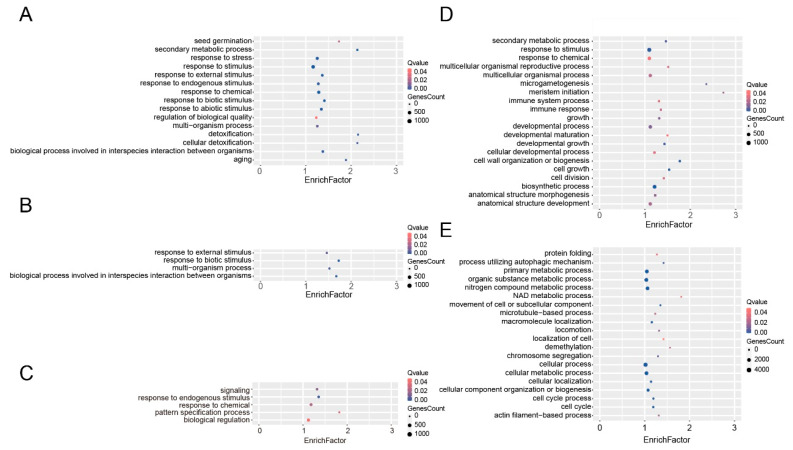
GO enrichment analysis of different types of duplicated genes for the A subgenome of *T. aestivum*. Panels (**A**–**E**) correspond to tandem duplication-derived genes, proximal duplication-derived genes, WGD-derived genes, dispersed duplication-derived genes, and non-duplicated genes, respectively.

**Figure 5 plants-12-03021-f005:**
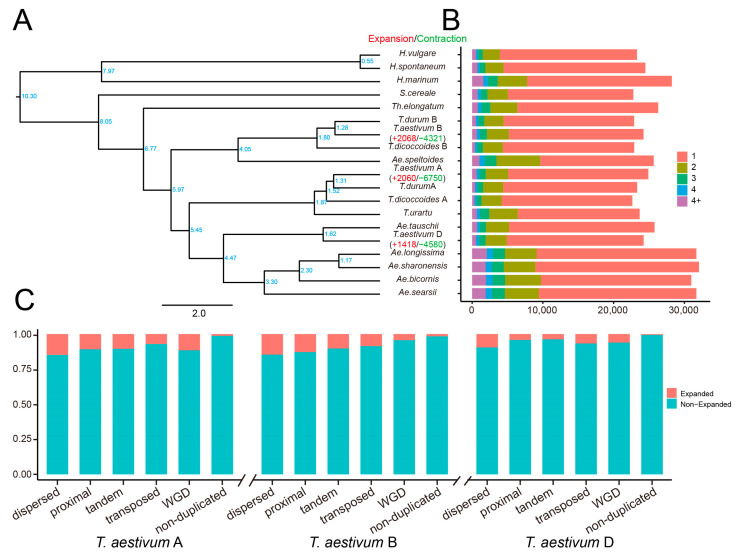
Gene duplication contributes to gene family expansion. (**A**). Gene family expansion and contraction analysis. The branches are labeled with numbers indicating the age of divergence, while the numbers following “plus” and “minus” represent the number of expanded and contracted gene families, respectively. (**B**) The distribution of copy numbers. (**C**) The contribution of gene duplication to gene family expansion.

**Figure 6 plants-12-03021-f006:**
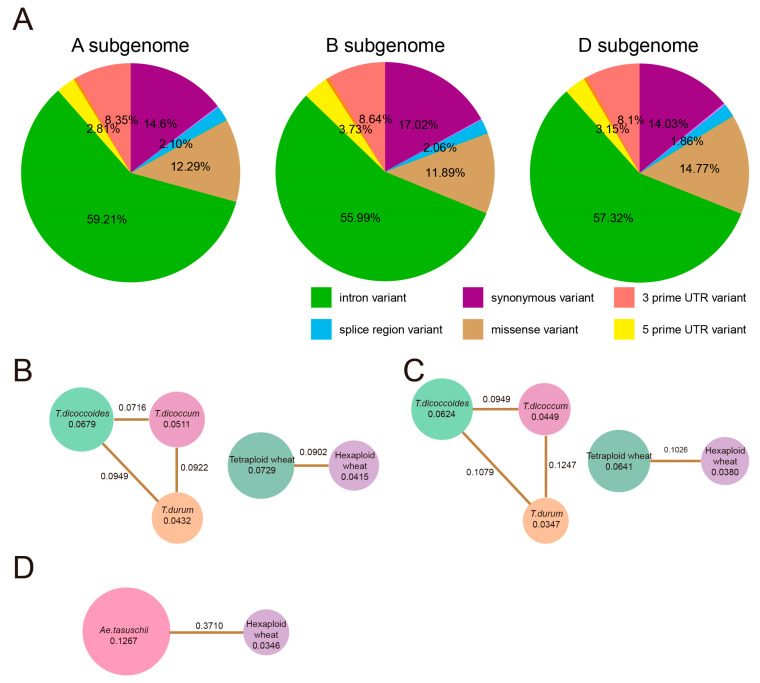
SNP distribution and genetic diversity analysis. (**A**). The distribution of SNPs in the A, B, and D subgenomes of *T. aestivum*, respectively. The pie chart indicates the gene ratio. The legend in the lower right presents the top six SNP types. (**B**–**D**). Statistics of nucleotide diversity and fixation index of *T. aestivum* A, B, and D subgenomes, respectively. The numbers inside the circles indicate genetic diversity, and the line between the two circles indicates the fixation index.

**Figure 7 plants-12-03021-f007:**
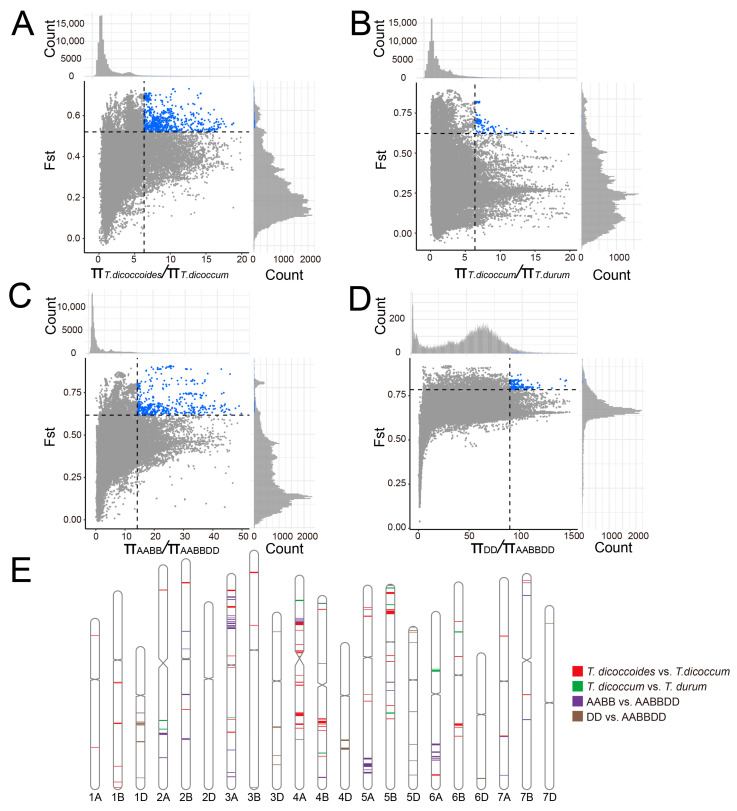
Identification of genomic regions exhibiting pronounced selective sweep signals. (**A**). Selective sweep signals between *T. dicoccoides* and *T. dicoccum*. (**B**). Selective sweep signals between *T. dicoccum* and *T. durum*. (**C**). Selective sweep signals between tetraploid wheat and hexaploid wheat. (**D**). Selective sweep signals between *Ae. tauschii* and Hexaploid wheat. (**E**). Ideogram showing the chromosome distribution of selective sweeps.

**Figure 8 plants-12-03021-f008:**
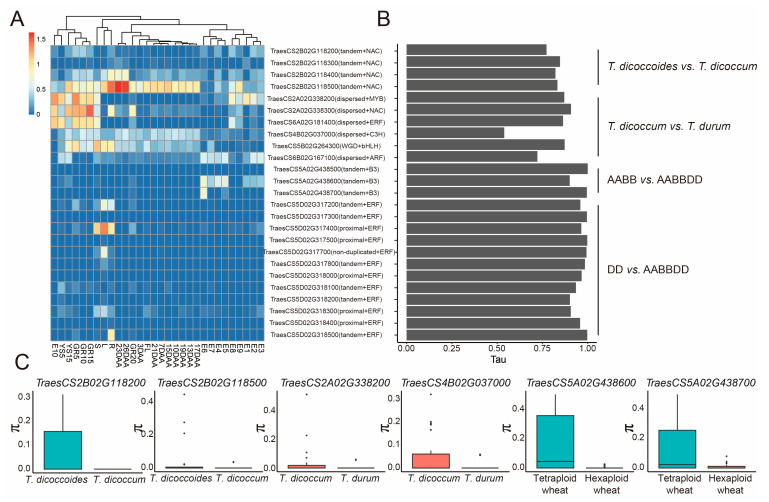
Hotspots of duplicated genes implemented in wheat domestication or hexaploidization. (**A**). Expression levels of candidate genes. 3DAA: Flag leaf, 3 days after anthesis; 7DAA: Flag leaf, 7 days after anthesis; 10DAA: Flag leaf, 10 days after anthesis; 13DAA: Flag leaf, 13 days after anthesis; 15DAA: Flag leaf, 15 days after anthesis; 17DAA: Flag leaf, 17 days after anthesis; 19DAA: Flag leaf, 19 days after anthesis; 21DAA: Flag leaf, 21 days after anthesis; 23DAA: Flag leaf, 23 days after anthesis; 26DAA: Flag leaf, 26 days after anthesis; E1: Embryo, two-cell embryo stage; E2: Embryo, pre-embryo stage; E3: Embryo, transition embryo stage; E4: Embryo, leaf early embryo stage; E5: Embryo, leaf middle embryo stage; E6: Embryo, leaf late embryo stage; E7: Embryo, mature embryo stage; E8: Endosperm, transition stage; E9: Endosperm, leaf late stage; E10: Seed coat (pericarp), Leaf early stage; FL: Flag leaf, booting stage (Zadoks 45); YS5: Young spike, booting stage (Zadoks 45); YS15: Spike, heading stages (Zadoks 53~54); L: Leaf, five-leaf-stage seedling (Zadoks 15); S: Stem, five-leaf-stage seedling (Zadoks 15); R: Root, five-leaf-stage seedling (Zadoks 15); GR5: Grains, 5 days post anthesis; GR10: Grains, 10 days post anthesis; GR15: Grains, 15 days post anthesis; GR20: Grains, 20 days post anthesis. (**B**). Tissue specificity of candidate genes. (**C**). The genetic bottleneck of candidate genes during wheat domestication or hexaploidization.

## Data Availability

All data are available within this publication and Appendix A.

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
