# Peer review of "Comparative Analysis Reveals Different Evolutionary Fates and Biological Functions in Wheat Duplicated Genes (Triticum aestivum L.)"

_plants, 2023, doi:10.3390/plants12173021_

Round 1

Reviewer 1 Report

In this manuscript, the duplicated genes in wheat were analyzed in detail by bioinformatics analysis and many innovative results were obtained. The results were well presented and the paper was well written. I think the manuscript could be accepted in current form.

Author Response

Dear Editor and Reviewers,

I hope you are doing well! Thank you for your comments on our manuscript entitled “Comparative analysis reveals different evolutionary fates and biological functions in wheat duplicated genes (Triticum aestivum L.)” (manuscript ID: plants-2560885). We have revised and resubmitted the manuscript accordingly. We apologize for any misdescription. Any specific suggestions on our manuscript are welcome. Once again, our sincere thanks.

All best regards

Xiuying Kong

Reviewer 2 Report

In this manuscript the authors identify duplicated genes within each wheat subgenome independently using the DupGen_Finder pipeline. They found that duplicated genes evolved faster than non-duplicated genes, and have lower expression levels and higher tissue specificity consistent with previous studies.  They classified the duplicated genes into different origins (e.g. tandem duplicates vs whole genome duplicates). They found that genes with different modes of duplication have different evolutionary rates and genetic diversity. They also identified hotspots associated with gene duplication.

Major comments:

(1) The use of the word “singleton” might be confusing because in wheat these genes still may have 3 homologous copies. Could these be termed “non-duplicated” genes?

(2) The entire article utilizes a significant number of statistical results, while some of the presentation of the particular data was not entirely clear. Such as:

Line 98-99: Please add the percentages of TD and DSD events.

Line 116: The authors mention “The Ka, Ks, and Ka/Ks values on the pericentromeric region were lower than those on the distal chromosome arm”. However from Figure S1 it is extremely hard to see any pattern in Ka, Ks or Ka/Ks. Can the authors present the data in a clearer way?

Line 150-151: Please use the statistical data to support this statement: “Singletons ranked among the positively selected genes as the most abundant type, followed by DSD- and TD-derived genes”.

(3) Some of the conclusions or statements do not seem justified:

Line 145: “No significant difference was observed for other duplicated genes and singletons” is not accurate. WGD- derived genes, the Ka/Ks values of the T. aestivum A subgenome were also significantly higher than that of the B subgenomes.

Line160-161: “These results demonstrated that the evolutionary rate might influence the T. aestivum gene structure” – These results are only correlations- the gene structure might be influencing the evolutionary rate, rather than the evolutionary rate influencing the gene structure.

And the same with Line201-202: “These results suggested that duplication mode might contribute more to shaping patterns of codon usage bias in T. aestivum”.

(4) Line 590: “For T. aestivum genes with one-to-many or many-to-many orthologous gene clusters, an average was computed.” Please can the authors explain how they calculated an average of multiple ortholog relationships?

(5) Line 364, Figure 6: It should be noted that the colour selection for SNP Types in A-C is not so appropriate; as bar charts, they do not serve the purpose of comparison effectively. In D-F, the meaning of the numbers and connecting lines is not indicated.

(6) All scripts used for analysis should be made available to allow reproducibility of this study.

Minor comments:

(7) Line 70: It would be better to present a clearly defined scientific question that the study aims to address.

(8) Line 91-92: References missing.

(9) Line 102-103: “DSD-derived genes and singletons displayed a scattered distribution along the chromosomes” From Figure 1B it looks like these classes are quite concentrated at the ends of the chromosome, perhaps not as strongly as other classes of duplication but the pattern is still noticeable.

(10) Line 107: Typo in Figure 1A – transposed

(11) Line 175: The caption for Figure 3A “*” and “**”is not clear.

(12) Supplemental Figures S4, S7 and S8 are hard to see because they are very long. Can the panels be re-arranged to be A4 in size?

(13) Figure S5B: Please centre-align the head of the “Intron length”.

(14) Supplementary Table15: It would be better to add a row for the total SNPs for genic regions which are mentioned in the article.

(15) Line 331-333: References.

(16) Line 404: What are E2, E8, GR5 and GR10 stages?

(17) Line 432: The designation "15DAA" (15 days after anthesis) should be explained as referring to leaves, in order to differentiate it from "GR15" (Grains at 15 days post anthesis).

(18) It would be better to replace acronyms with full name of duplication types throughout the manuscript to improve readability and match the figures (WGD is ok to keep). 

Author Response

Dear Editor and Reviewers,

I hope everything is going well for you! Thank you for your consideration in providing the opportunity to refine the quality of our manuscript entitled “Comparative analysis reveals different evolutionary fates and biological functions in wheat duplicated genes (Triticum aestivum L.)” (manuscript ID: plants-2560885). We would also like to thank the reviewers for their comments on the manuscript, which were made more valuable by their thoughtful comments and suggestions.

Our point-to-point responses to reviewers’ comments are provided in this cover letter file (Please see the attachment). Hope the revised submission meets the standard of Plants and could be accepted for publication.

All best regards

Xiuying Kong

Reviewer 3 Report

The manuscript entitled “Comparative analysis reveals different evolutionary fates and biological functions in wheat duplicated genes (Triticum aestivum L.)” reported the wheat duplicated genes from whole genome level. This study clearly showed the types and numbers of duplicated genes, evolution rate, divergence in codon usage, expression level, gene family, domestication and polyploidization as well as hot spot. This work provided new information for understanding of wheat duplicated genes.

Author Response

Dear Editor and Reviewers,

Hope this email finds you well. We appreciate the reviewers very much for these valuable comments on our manuscript entitled “Comparative analysis reveals different evolutionary fates and biological functions in wheat duplicated genes (Triticum aestivum L.)” (manuscript ID: plants-2560885). We hope to have addressed these concerns to the reviewers’ satisfaction, and any additional suggestions for improving our manuscript are welcomed. Sincere thanks again.

All best regards

Xiuying Kong